# Bio-Functionalized Ultra-Thin, Large-Area and Waterproof Silicone Membranes for Biomechanical Cellular Loading and Compliance Experiments

**DOI:** 10.3390/polym14112213

**Published:** 2022-05-30

**Authors:** Karya Uysal, Till Creutz, Ipek Seda Firat, Gerhard M. Artmann, Nicole Teusch, Aysegül Temiz Artmann

**Affiliations:** 1Institute for Bioengineering, Campus Juelich, University of Applied Sciences Aachen, Heinrich-Mussmannstr. 1, 52428 Juelich, Germany; uysal@fh-aachen.de (K.U.); till.creutz@fh-aachen.de (T.C.); firat@fh-aachen.de (I.S.F.); artmannmg@gmail.com (G.M.A.); 2Institute for Pharmaceutical Biology and Biotechnology, Heinrich-Heine University Düsseldorf, Universitätsstr. 1/Geb. 26.23, 40225 Dusseldorf, Germany; nicole.teusch@hhu.de

**Keywords:** bio-functionalization, flexible membrane, polydimethylsiloxane (PDMS), polymer functionalization, large-area membrane, spectral reflectance measurement, elastic modulus, stress distribution, membrane technology, biomedical applications

## Abstract

Biocompatibility, flexibility and durability make polydimethylsiloxane (PDMS) membranes top candidates in biomedical applications. CellDrum technology uses large area, <10 µm thin membranes as mechanical stress sensors of thin cell layers. For this to be successful, the properties (thickness, temperature, dust, wrinkles, etc.) must be precisely controlled. The following parameters of membrane fabrication by means of the Floating-on-Water (FoW) method were investigated: (1) PDMS volume, (2) ambient temperature, (3) membrane deflection and (4) membrane mechanical compliance. Significant differences were found between all PDMS volumes and thicknesses tested (*p* < 0.01). They also differed from the calculated values. At room temperatures between 22 and 26 °C, significant differences in average thickness values were found, as well as a continuous decrease in thicknesses within a 4 °C temperature elevation. No correlation was found between the membrane thickness groups (between 3–4 µm) in terms of deflection and compliance. We successfully present a fabrication method for thin bio-functionalized membranes in conjunction with a four-step quality management system. The results highlight the importance of tight regulation of production parameters through quality control. The use of membranes described here could also become the basis for material testing on thin, viscous layers such as polymers, dyes and adhesives, which goes far beyond biological applications.

## 1. Introduction

Synthetic membranes have been used in many major applications for about one hundred years. Their manufacturing processes have been developed rapidly since then, as their application horizon appeared very wide [1]. However, their functional usages did not exceed applications such as the separation of substances or the purification of water. It was not until the mid-1940s that they had attracted the interest of biomedical researchers. Membranes were subsequently developed mainly for hemodialysis, paving the way for membrane oxygenators, new drug delivery methods and lab-on-a-chip systems [2,3].

To expand the variety of applications in life science research, scientists began to chemically/biologically functionalize the surfaces of polymer membranes [4]. This led to better functioning membranes and/or membranes with tunable properties such as biocompatibility, mechanical resilience, or their degradability. Only suitably modified membrane surfaces are useful in biomedical research, and giving them greater physiological relevance. Membranes no longer form a simple barrier or filter for biological material but also allow, among other things, the cultivation of cells of any kind on their surface [5,6].

In the past decades, PDMS membranes became the focus of interest. They were mainly used to study the mechanical properties of single cells or cell layers in vitro, applying compressive or longitudinal tensile loads. Some research groups developed measurement techniques that allowed the calculation of cell tensile forces, cell deformation and stiffness by modifying commercially available instruments (StrexCell, San Diego, CA, USA, FlexCell, Burlington, NC, USA and others) [7,8,9,10]. These companies offered mechanical traction applications with uniaxial and biaxial stretching options, as well as compression systems aimed at mimicking the complex physiological and biomechanical nature of cell and tissue mechanics in living nature [11]. This ultimately enabled basic biological and medical researchers to open up the new field of cell and tissue biomechanics science. The cell seeding area in such applications of membrane arrays varied from 1 to 10 cm^2^ with membrane thicknesses of 100–500 μm. These were offered either “untreated” or “functionalized” for improved cell adhesion, and in the stiffness modulus range of 1–60 kPa. Even though few disposable membranes were commercially offered to be used in combination with these devices, some systems still required researchers to produce their own PDMS membranes followed by surface functionalization, leaving space for the optimization of membrane production along with post-production quality control [9].

Most known methods to produce PDMS membranes are spin-coating [12], casting [13] and soft lithography [14]. Spin-coating is based on the spinning of an elastomer mixture at high speed, leading to PDMS coating of the underlying surface. Casting is performed by the spreading of the same mixture by a so-called casting knife [13]. Finally, soft lithography used a simple mold casting process creating PDMS membranes/scaffolds in the required thickness, also allowing patterning on the surface [14]. A method for fabricating very thin, large-area membranes published in 2019 has been termed the floating-on-water (FoW) process by those authors [15]. There is one major drawback in ALL these methods: They were NOT suitable for the fabrication of <10 µm thin, large-area membranes to be used as sensors for measuring mechanical stresses of a few N/m^2^ as they occur in (cell) layers only a few µm thin. The requirements for such membranes, such as firm and uniform adhesion to the adhesive surface at the edges, avoidance of even the slightest membrane wrinkles and uniform mechanical preload and preload distribution, are extremely high if we expect even a remotely reproducible application for mechanical (cell)-tension measurements. The fabrication of membranes floating on a water surface was proposed and patented by Artmann as early as 2000 [16,17,18,19].

Using such sophisticated membranes, a cell force measurement method—CellDrum technology—was developed. It was used practically for the first time in the dissertation by Trzewik et al. [20]. Since then, the fabrication method became the basis of many biomedical applications of this group. Important applications were the measurement of cell force of fibroblasts [20,21] and the beating force and frequency measurement of hIPS cardiac myocytes as well as the contraction force measurements of smooth muscle cells [22,23,24].

The aim of this work was to produce CellDrum membranes with a predetermined and reproducible thickness between 2–10 μm and an area of up to 2.1 cm^2^. They should be waterproof, transparent, chemically/biologically functionalizable and suitable for cell culture. Production of such ultra-thin membranes allow for high membrane extensibility, which makes their stretching sensitive enough to small changes in lateral contraction forces of cell layers adhering to them on top. In addition, thin membranes exhibit improved gas permeability. With their high extensibility as well as their good gas permeability, such (waterproof) membranes provide an environment that is close to the in vivo situation of cells. Thus, cell force measurements and their changes in physiological or pharmacological experiments are better adapted to those in the human body. In contrast, hard substrates of cell culture dishes lead to largely non-physiological biological and biomechanical cell behavior. Since these CellDrum membranes serve as sensors for extremely low mechanical stresses as they occur in cultured cell layers, their mechanical properties must be precisely known and the reproducibility of CellDrum-based measurements must be high [16,17,18,19,25]. These highly flexible membranes allow in addition to mechanical tension measurements external mechanical stresses to be introduced into the cell layer cultured on top of the membranes [26]. Flexible membranes, in contrast to the hard surfaces of cell culture flasks, offer a biomechanically relevant in vivo-like soft surface for cells. The measurement results obtained with CellDrum membranes thus have a higher physiological relevance.

We hereby present the membrane production procedure as well as functionalization method, along with the standard three-step quality control protocol for highly precise mechanical cell tension measurements based on the CellDrum technology. These membranes mounted on so-called CellDrums [21,25] were developed in the past together with a Cell Force Analyzer instrument (CFA_refer_) [27]. They are used for research in the medical and pharmacological field, in tissue engineering, drug testing and wound healing studies. Thanks to their biological-physiological properties, non-toxicity and flexibility as well as low production costs, they are ideally suited for this purpose.

## 2. Materials and Methods

### 2.1. Membrane Fabrication

The PDMS elastomer kit (Sylgard 184, Dow Corning, Midland, MI, USA) was utilized as recommended by the manufacturer. The curing base and the curing agent were mixed in a ratio of 10:1 (*w*/*w*), resulting in a mixture viscosity of 3500 cP. For degassing, the tube containing the elastomer mixture was centrifuged at 3200 rpm for 2 min (Heraus Biofuge Primo), and this step was repeated twice to ensure the homogenous mixing of the two components. For the production of the thin membrane (<10 µm), FoW method was chosen [15]. Deionized water, degassed by a desiccator, was used as the substrate for the PDMS, and was pipetted into sterile 6-well plates. Dimensions of a single well are as follows; 35.3 × 15.9 mm (W × D), with an area of 9.6 cm^2^. The PDMS was transferred onto the water using a positive displacement pipette to prevent air bubbles from forming during pipetting. Due to the higher surface tension of water in contrast to PDMS, the mixture spreads over the liquid surface once in contact. To cure the membranes, heat lamps were used above the wells (Figure 1B), and the wells were placed accordingly to ensure even heat distribution. The heat curing was performed at room temperature (24 °C) for 35 min, and the final measured water temperature stands between 55–60 °C. Cured PDMS membranes were transferred onto the CellDrum applicator with the help of a mechanical aid. The produced membrane area is four to five times bigger than the CellDrum area (Figure 1A,C). The mechanical aid ensures that all membranes are transferred onto the applicators centrally.

The first experiment regarding membrane thickness and production parameters was performed by pipetting different PDMS volumes onto the water surface in each well of a 6-well plate. Three batches of membranes were produced with 4-, 8- and 10 µL (max. pipette volume) of PDMS and thicknesses were measured. All three batches were produced in the same day, though not simultaneously. The second experiment was performed using one constant PDMS volume (4.15 µL) at three different temperature settings (22–24–26 °C). Temperature regulation for the temperature dependency of membrane thickness trials were carried out with a heater system, and a digital room thermometer was used to track the temperature. All experimental conditions can be seen in Table 1.

### 2.2. Membrane Functionalization

To create a hydrophilic surface for the biological coating, firstly a two-step wet-chemical functionalization protocol composed of an oxidation step and a silanization step was used [28,29]. All used solutions were prepared freshly prior to membrane functionalization. The oxidation solution was prepared following the procedure by Sui et al. as H_2_O/H_2_O_2_ (30%)/HCl (37%) were mixed in a 5:1:1 volume ratio [28]. Membranes were exposed to the solution for 30 min, washed and the same step was repeated two more times, without waiting time between washes. Next silanization step was performed with a 2% solution of trimethoxy [2-(7-oxabicyclo[4.1.0]hept-3-yl)ethyl]silane in an Isopropanol/Water (19:1) mixture with a pH of 4.9–5.0, regulated using acetic acid and was kept on the membrane surface for 5 min [29]. The silanization solution is discarded and the membrane is washed using deionized water, and is autoclaved at 121 °C, 0.2 MPa for 20 min to stabilize the previous silanization step.

For the biological functionalization, fibronectin from bovine blood plasma (Sigma Aldrich Chemie, Taufkirchen, Germany) was chosen as matrix coating. The ready-to-use solution was diluted down to 1% (*v*/*v*) concentration using a 50 mM, 2-(N-morpholino)ethanesulfonic acid (MES) buffer made with autoclaved deionized water [30]. The buffer pH was set to 6.1 using 1 M NaOH solution and sterilized using a 0.22 µm filter. Finally, the fibronectin solution is removed after 24 h incubation time and the membranes are kept refrigerated with phosphate buffer saline solution (PBS) until use.

The main focus of this paper is the investigation of membrane mechanical properties and description of a standardized production procedure as well as quality management. Therefore, to determine in this study only membranes with chemical functionalization were used.

### 2.3. Post Fabrication Quality Control

The semi-automatic laboratory-measuring device, the CellForce Analyzer, or CFA_refer_ for short, was developed for measuring mechanical stresses based on the CellDrum technique in cultivated cell layers [25]. The device as well as quality controlled CellDrums have been commercially available since 2022 (npi electronics GmbH, Tamm, Germany). It was initially intended for applications in biological and clinical research. The CFA_refer_ is a multi-channel device that can measure six CellDrums under controlled conditions. The membrane manufacturing processes discussed here as well as the necessary quality management represent an important part of the developmental work. The membranes in the CFA_refer_ are mounted on six individual CellDrum holders. They must meet the usual requirements for sensors in terms of accuracy, reproducibility and other properties. In particular, their sensitivity to mechanical loads must be known. The quality control described here is dedicated to the goal of manufacturing the CellDrums in such a way that they deviate from each other by less than three percent in their mechanical sensitivity. Only in this way is the accurate measurement of cell forces and their changes even conceivable.

### 2.4. Four-Step Quality Control Procedure

#### 2.4.1. Visual Examination

All membranes used for the experiments must pass a preliminary visual examination before they are tested further. Membranes must not show any wrinkle formation on the attachment sites and must be free of holes (Appendix A) or dust particle inclusions over the whole surface to prevent non-uniform stress distribution as well as any leakage of fluids. Membranes that do not meet these requirements are discarded and no longer further reducing unnecessary work and shortening the overall duration of quality control testing.

#### 2.4.2. Membrane Thickness Measurement

To measure the membrane thickness of the thin PDMS films, Filmetrics F20 (KLA Instruments, San Diego, CA, USA) spectrometer was used. This device operates with high precision according to the interferometer principle. It is sensitive to jumps in the refractive index at the two surfaces of the membrane. It measures the light reflectance of the surface over a range of wavelengths to determine the film thickness. Once the measurement is completed, the data are analyzed by the comparison to a calculated reflectance spectrum, which is a function of the given material’s refractive index. The setup also allows thickness determination of multiple layers through a “recipe” system, where the user must define the structural order of different materials. In the measurement of this thin PDMS membrane, the used recipe was “Air-PDMS-Air” as the thin film is free-standing, therefore not connected to any other material and is simply surrounded by air. The calibration of the device can be controlled with a thickness standard, also provided by the company Filmetrics, and the goodness of fit (GOF) can be regulated by adjusting the height of the lens which supplies the system with the light through fiber optic cables. Special holders were constructed and 3D-printed to ensure that the thickness measurement was performed at the same seven spots for each membrane (Figure 2). Eccentric aid allows the CellDrum to be rotated via a revolver which provides a 60° spacing between each measurement point. In this study, only the central measurement results are presented.

Membrane thickness depends on the control of multiple parameters such as elastomer volume, mixing ratio, curing temperature and initial PDMS and water temperature before curing [15,31]. For the investigation of thickness dependency on temperature, three different room temperatures were applied for testing (22–24–26 °C), using PDMS and water that were also stored in the same room before production, to keep the parameters as uniform as possible. Another parameter used to investigate the dependency of the final membrane thickness was the PDMS volume. For this purpose, three different PDMS volumes (Table 1) were tested and compared between each other.

#### 2.4.3. Membrane Deflection Measurement

Membrane deflection was measured with a laser triangulation sensor (LK-031, Keyence Deutschland GmbH, Neu-Isenburg, Germany) to determine the thickness dependency of the mechanical properties of the membranes. The custom graphical user interface software was programmed using LabView^®^ (LabVIEW 17.0.1f3, LabView, National Instruments Corporation, Austin, TX, USA), which was designed to read the sensor signals and give numerical outputs for the measured values.

CellDrum applicators are cylindrical plastic rings. They are covered from the bottom with the PDMS membranes making them a “CellDrum”. CellDrums are placed into a socket for light deflection measurement (Figure 3). For membrane deflection tests, 500 µL of PBS was pipetted onto the membrane surface as mechanical load, leading to membrane displacement. Prior to the measurements, fine talcum powder was used to create non-transparent spots on the center of the membranes with the help of a soft stamp which allows the laser signal to be read by the internal charge-couple device (CCD) sensor. A blank measurement (membrane without load) was taken for calibration before every deflection measurement. In case of use with cell monolayers, only a few representative membranes from one batch would be chosen for deflection measurement, to prevent contamination caused by talcum powder. Based on these data, a comparative prediction about the elasticity of the membranes was made.

#### 2.4.4. Membrane Compliance Measurement

Mechanical compliance is a measure of flexibility of a material. The more compliant a material is, the more it gives in under load. Usually, the compliance of the CellDrum membrane is measured with the CFA_refer_ System. For that, the instrument slowly increases the pressure under the membrane deflecting it upwards. The pressurizing affects the strain within the membrane which can be indirectly determined by measuring the pressure under the membrane and the membrane’s deflection. The compliance parameter was introduced in particular for the intuitive understanding of the measured data in biomedical studies [24].
Strain ε = ΔLength/Length(1)

When plotting a graph of deflection over pressure, the measured data represent a sigmoid curve crossing the line of minimum strain. At the zero crossing point the compliance value is then acquired by conversion of the pressure and deflection value to receive the ratio of the change of strain to the change of stress, as depicted in the following formula:Compliance C = ΔStrain/ΔPressure(2)

The measurement of CellDrum compliance or mechanical stress according to Equations (1) and (2) is also used to determine the compliance of a cell layer cultured on the PDMS membrane (cell-membrane-composite layer) [11,24,25]. Since the cultured cells on the CellDrum membrane are tightly adhered to the surface, the cells contribute to the bulk mechanical membrane properties. The contribution of the mechanical lateral stress of the cell layer to the measured total compliance or the total mechanical stress, respectively, of the PDMS membrane-cell layer—the “compound”—is only measurable if the membranes are sufficiently thin as described above. Furthermore, two boundary conditions must be met: (1) the membrane deflection must be able to be measured very accurately under well-controlled boundary conditions, and (2) each individual membrane must be accurately measured in the absence of cells. These data must be included in the evaluation of the mechanical stress or compliance measured with cells and the same membrane.

CFA_refer_ comes with a desktop user interface program. Once a user has inserted the CellDrums with cells into the device, the pre-measured membrane parameters must be typed in. This is of high importance for the precision of the measurement. The user can set up six CellDrum measurements at once. Moreover, an automatic measurement repetition at a set rate is possible. The CellDrums are measured sequentially and can be saved individually for later processing. Finally, the user can select data to create a comparative graph file with the program. Compared to former prototypes, CFA_refer_ is easier to use, has improved measurement-time window and can measure an exemplary batch of six CellDrums with five measurement sets in about 15 min. Post processing is equally easy and intuitive.

#### 2.4.5. Statistical Analysis

The statistical analysis was performed using IBM’s SPSS Statistical Analysis. Normality tests were performed for all groups with a Shapiro–Wilk test due to smaller group sizes (*n* < 50). For all experimental setups one-way ANOVA tests were performed. For groups that show overall significant differences, further post hoc tests were carried out (Tukey’s HSD) to determine which exact groups differed between each other. For correlation tests, Microsoft Excel data analysis tool was used with Pearson’s correlation coefficient.

## 3. Results

### 3.1. Dependency of Membrane Thickness on PDMS Volume and Ambient Room Temperature

The CellDrum technology used since 2002 was adapted to the available laboratory environment along minor changes such as the size of the used dish, heating method and curing time. Through these changes, the used volumes of water and PDMS were regulated accordingly. Two parameters were investigated in connection to membrane thickness and fabrication parameters: (1) the used PDMS volume and (2) the ambient room temperature at which PDMS spreads over the water surface (Table 1). When the PDMS volume is pipetted onto the water surface, the elastomer spreads out in a circular pattern, which can be followed briefly after pipetting in the reflected light. After just a few seconds, the PDMS liquid film on the water surface is difficult to detect due to its transparency. The distance between the PDMS drop that is formed at the tip of the pipette and the liquid surface was kept as small as possible to prevent any bouncing and straying of the elastomer on water, which can lead to non-uniform thickness spots on the membrane. To keep environmental effects to a minimum, all experiments were performed in the same room. Due to the weak bonding between the water surface and PDMS, the cured membranes were easily transferred onto the CellDrum applicators as free standing membranes (see Figure 1A,C).

Firstly, three different volumes of PDMS were tested for membrane production at room temperature. Changing the PDMS volume is the most direct way to influence the membrane thickness, thus proportional volumes of 4-, 8- and 10 µL were examined and the correlation between groups were analyzed. Using the simple mathematical formula to calculate membrane thickness as it was also used by Kim et al., it is expected that the values would linearly increase with the changing volume of PDMS [15]. However, the measured thicknesses did not completely match the calculated values as shown in Table 2 below.

Mean membrane thickness for the 4 µL group (Group 1) was 3.47 ± 0.23 µm (*n* = 11). The volume was then doubled, resulting in a mean of 7.49 ± 0.26 µm (*n* = 11) for group 2. The expected theoretical 100% increase should result in a membrane thickness of 6.94 µm. However, the measured increase was 116% which equals to a 1.16-fold increase. For group 3, 10 µL of PDMS was used. The mean thickness for this batch was 9.43 ± 0.63 µm (*n* = 11). The expected fold increases between group 1–3 was 1.50, and between group 2–3, it was calculated as 0.25. Measured fold increases between the same groups were 1.72 and 0.26 respectively. The nominal (calculated) and real (measured) fold increase values between groups 2 and 3 match perfectly with the values 0.26 and 0.25. Under ideal circumstances, a fold increase of 1.5 was expected between groups 1 and 3, as the used volume is 2.5 times higher than group 1. However, the measured fold increase was 1.72, showing once again a thicker membrane.

These results could be because the PDMS drop does not spread exactly circular and/or cures non-uniformly, causing the central thickness of the membrane to be thicker than expected. Another reason for uneven spreading of the elastomer could be related to the lack of exact positioning of the PDMS drop in the center. Environmental disturbances causing air flow on the water surface might contribute as well. Thus, the membrane curing procedure location should be isolated from the room ventilation. Using the holder with six circular measurement points (see Figure 2), variations in thickness throughout membrane surfaces were detected, supporting the first hypothesis on non-uniform PDMS spreading. However, these variations were below 10%, keeping the thickness distribution in a confidence interval of 90%, and therefore were not reported in this paper in further detail. All three groups were also compared to each other in a one-way ANOVA test to show significance, a proof of non-coincidental results (Figure 4). The temperature during production was kept within ±0.5 °C. Temperature deviations result in PDMS viscosity changes causing thickness variations between the three groups. Therefore, effects of temperature were tested in the next trial.

Next trial concerning membrane thickness was performed using the same PDMS volume (4.15 µL) at different ambient room temperatures during the spreading of PDMS elastomer on the water surface. Chosen values were 22, 24 and 26 °C, all of which are common temperatures of laboratory environments across seasons. Temperature regulation was controlled by the same means for every trial, and did not vary during fabrication preparation. Group sizes were identical for all groups, *n* = 14. All three batches tested positive for normal distribution before the one-way ANOVA test was performed (Figure 5). The first temperature group at 22 °C (Group 1) had the highest average membrane thickness of 4.09 ± 0.40 µm. The second group at 24 °C (Group 2) was 3.72 ± 0.30 µm thick and the thickness of the third group at 26 °C (Group 3) was 3.68 ± 0.35 µm (Table 3).

The viscosity of liquids, including liquids spreading on water, increases with decreasing ambient temperature, which explains the observed higher central thickness of the membranes at lower temperatures [32]. At lower temperatures, PDMS over water surface also spreads visibly slower as compared to higher temperatures. Despite the small range of examined temperatures, significant differences were seen. Between groups 1 and 3 there was a 4 °C ambient temperature difference only. However, the thickness decreased significantly (One-way ANOVA, α = 0.01) by about 10% (Table 3, Figure 5). Thus, the effect of ambient room temperature must be considered technologically for CellDrum production and their use with the CFArefer. Consequently, PDMS should be pipetted at constant ambient temperature in a closed container, which also guarantees dust-free air.

### 3.2. Membrane Functionalization

Functionalization methods of ultra-thin CellDrum membranes were published previously, as this measurement technique has been used for measurements with cell layers since 2002 [25,28,29,30]. As there are no universal and strictly defined “measurements” of biocompatibility, in this paper, the term “biocompatibility” was defined as providing cells with a healthy cell attachment, survival and proliferation environment on the CellDrum membrane. Aforementioned studies successfully proved the functionality and the biocompatibility of this chemically and biologically modified PDMS surface. The cells were able to attach and proliferate on the membrane over long time studies of two weeks without any sign of cellular apoptosis, while also responding to external stimulation and/or medical substance additions (Figure 6) [21,22,23,24].

Increased wettability was measured by a surface tension investigation via critical surface energy detection (γ_c_ in mJ/m^2^) in a previous study performed in the same laboratory. Uncoated CellDrum membrane surface tension was examined in comparison to plasma etched and fibronectin coated membranes. The results of the study proved the functionality of the membrane coating procedure, with improved surface wettability [20].

Along with the published data, many studies that are yet to be published were performed in our laboratory using the CellDrum technology, in which no problems were encountered with the cell cultivation over the membrane surface, once again proving the efficiency of the chemical functionalization as well as the biological fibronectin coating.

### 3.3. Interdependency of Membrane Thickness and Membrane Stiffness

#### 3.3.1. Membrane Deflection

For deflection measurements, two test setups were used. To create membrane displacement, both measurement setups used 500 µL PBS load. Firstly, the deflection of the membrane was measured in a batch of chemically functionalized membranes (*n* = 35) produced simultaneously in the same environment. Even though the fabrication and quality control procedures of the membranes were standardized, due to the manual production minimal parameter fluctuations were to expect. Membrane transfer via the mechanical aid onto the CellDrum applicator (Figure 1A) provides the membrane with an initial pretension within the membrane, that cannot be manually regulated. Different pretensions on membranes lead to different deflection values despite using the same forces/loads. However, this can be limited to a certain degree by automatization of the production process inside of a manufacturing container. Thicknesses of these membranes ranged between 3.28–3.98 µm. The relationship between thickness and stiffness was statistically analyzed by a linear regression test (Pearson’s Function) after proving normal data distribution using IBM SPSS. The function gives values between “−1” to “1”, where both values indicate a strong correlation, either negatively or positively, while a “0” value shows no relationship between two parameters [32]. A one-way ANOVA test further proved the similarity and comparability of the data with a non-significant result at α = 0.05 (Figure 7).

We investigated the correlation between the deflections of the CellDrum membranes of different membrane thickness ranges and the measured deflection upon hydrostatic loading of 500 µL of PBS on the CellDrum membranes (inner diameter 16 mm) (Figure 7, Table 4). Data showed a Pearson correlation coefficient of 0.10 reflecting that there was no correlation found within the entire membrane thickness range analyzed (3.28–3.98 µm). This may be due to the fact that the chosen thickness range is very narrow. However, the range investigated is the most commonly used thickness range for the 16 mm in diameter CellDrums.

The second test series concerning membrane deflection was carried out with the same membranes that were used above in Section 3.1 (Table 2). The deflection measurements of the 3 PDMS volume groups (4, 8 and 10 µL) with *n* = 11 were repeated three-times and compared using a one-way ANOVA test. Almost a doubling of the deflection values was seen between 4- and 8 µL as well as 4- and 10 µL volume groups, which can be further visualized in Figure 8. While an average deflection of 844 ± 52 µm is shown in group 1, group 2 and 3 revealed deflection values of 439 ± 19 µm and 443 ± 24 µm respectively (Table 5). With a one-way ANOVA test, high significant differences between the deflection values were shown (*p* < 0.01) between group 1 vs. group 2 and group 1 vs. group 3. However, no significant difference between group 2 and group 3 was realized, which could be due to the smaller volume and thickness difference between the membranes of each group, compared to group 1.

The one-way ANOVA test showed no significant difference between the five thickness groups that were shown in the first deflection trial (Figure 7). However, as the thickness range was increased, highly significant differences were revealed between different thickness groups at α = 0.01 (Figure 8).

In 2009, Liu et al. investigated the thickness dependency of PDMS membranes’ mechanical properties, produced by spin coating method. These characteristics were previously considered to be dimensionally independent. His research group revealed as expected that the Young´s modulus as well as the general mechanical strength of the PDMS membranes to be thickness dependent. They defined ranges for different mechanical behaviors known as bulk behavior and dimension dependent behavior (<200 µm). This transition in behavior is found to be correlating to a thickness dependent reorganization of the polymer chains to create more resilient cross-linked networks, during membrane production [33]. All membranes used in our study were significantly below the transition value of 200 µm which fit the thickness dependent behavior range. However, there was no linearity or pattern found between the two investigated parameters in the thickness range of 3.28–3.98 µm. The use of such small thickness range with might be the reason why no significant results were obtained for the first deflection trial (Figure 7). The thinnest membrane used in the study by Liu was 30 µm, while ours is roughly about 1/10th of that value. That could be the initial reason to the lack of correlation between membrane thickness and deflection values. Despite that, the second deflection trial (Figure 8) with the bigger thickness range (3.24–10.06 µm) showed significant dependency of measured deflection values and the membrane thickness. A decreasing membrane deflection was detected with increasing membrane thickness. The results prove that wider ranges of membrane thickness must be chosen for investigation for ultra-thin membranes, supporting the hypothesis of Liu et al. on dimension dependent behavior. Another study from 2014 by Johnston et al. studied the dependency of membrane mechanical properties such as Young’s modulus, ultimate tensile strength (UTS), ultimate compressive strength (UCS) and compressive modulus on curing temperature. He reported significant differences for these parameters in a temperature range of 25–200 °C [34]. In the study, 200 µm thick membranes were used. Membranes above 200 µm thickness are believed to show bulk behavior. However, Johnston et al. revealed an increase in the Young´s Modulus (MPa) of the PDMS membrane with increasing curing temperature. Membrane mechanical properties were also proven to be dependent on curing temperature. Therefore, it is highly likely that these properties could also be influenced by ambient room temperature as reported in our temperature dependent membrane thickness results (Figure 5, see Section 3.1). Even though ambient room temperature was kept as stable as possible for the production of all membranes for the deflection measurements, slight fluctuations (±0.5 °C) could have occurred. These fluctuations could have had an additional effect on the measured deflection values.

To further show how ultra-thin membranes can differ in mechanical characteristics under identical production parameters, an additional calculation was performed. Five membranes with the same thicknesses from the first deflection trial (*n* = 35) were chosen and their measured deflection values were examined. These membranes had an average thickness of 3.648 ± 0.004 µm. Deflection values for each membrane were as follows; 916 µm (1), 1378 µm (2), 939 µm (3), 569 µm (4) and 1008 µm (5), respectively. These results prove once again that the membrane thickness is not the only parameter which affects mechanical membrane properties. As mentioned before, a pretension is provided to the membrane as it is stretched over the CellDrum applicators (Figure 1A). This pretension is a determinant factor of membrane deflection as it directly influences how the membrane reacts to external additional loads on top of the surface. The membrane transfer from the mechanical aid onto the applicators is a manual procedure. Therefore, deviations in the intensity of this pretension are expected. The only way to minimize pretension deviations between membranes, the production system and/or at least the membrane transfer after curing, must be automatized. The automatization is expected to decrease or completely prevent unequal distribution of forces during the mechanical transfer of the membrane, minimizing possible inconsistencies.

#### 3.3.2. Membrane Compliance

To overcome possible systemic errors from the deflection trials (see Section 3.3.1), a different test setup was used. A sample group of 30 CellDrums was investigated for their compliance in the CFA_refer_ measurement device. The results were separated into four thickness groups with seven to nine samples each, with the help of a histogram. The groups average compliance value was calculated (Table 6). A Shapiro–Wilk test proved that all populations were normally distributed, so a one-way ANOVA test could be performed. The test showed no significant difference between the groups, when using a significance level of α = 0.05 (Figure 9).

The results indicate that there might be little interdependence of membrane thickness and membrane compliance for membranes thinner than four micrometers. This is contradicting at first glance, but considering the very low thickness of the membranes, it becomes apparent that other factors might overlay the mechanical behavior at this point. Liu et al. found in their experiments that membrane thickness values below a threshold of 200 µm tend to shift from bulk behavior to dimension dependent behavior. This might be due to a thickness dependent reorganization of the PDMS polymer coils [33]. However, their tests were performed at much coarser thickness intervals and never came close to our membrane thicknesses. Rather, production circumstances such as curing temperature might have a considerable large influence on the mechanical properties, as pointed out by Johnston et al. [34]. These factors might even shadow the influence of thickness on these membranes. The pretension of the membranes is not measured separately in this study. However, in addition to other parameters, it decisively determines the deflection of the membranes under otherwise identical conditions.

All speculations aside, it becomes apparent that there is little known when it comes to the properties of ultra-thin PDMS membranes, increasing the need of a strict quality control and further investigation of the matter.

### 3.4. Membrane Stiffness Comparison to In Vivo

The compliance of 30 measured CellDrum membranes obtained by the CFA_refer_ device averaged a value of 5.3 × 10^6^ Pa^−1^, which equates to a stiffness value of approximately 190 kPa. In comparison, tests of the stiffness of PDMS of the same blend claimed to have shown values of 1.32–2.97 MPa [35]. This indicates that the here used production methods might have lowered the materials stiffness substantially. Nonetheless, the results should be taken with a grain of salt, since there seems to be a rather large variation of stiffness of PDMS depending on the production and testing method. In addition, the mentioned CFA_refer_ measuring device is still in a constantly evolving stage and so the measurement result may change slightly. As the CellDrum membranes are also meant to be used as culture vessels, these should be compared to commonly used tissue culture plates/flasks made of Polystyrene, and to tissue types found in the human body. Skeletal muscle and cartilage have an elasticity of 12 and 42 kPa respectively [36,37]. That makes the PDMS membranes considerably stiffer than these tissues. However, when compared to Polystyrene flasks that have an elasticity of 10 MPa, a PDMS membrane becomes desirable again. Furthermore, the presented PDMS membranes showed to be easily functionalized with protein coatings. If low surface elasticity is demanded, coating the membrane with a low elasticity hydrogel, for instance, could be easily fulfilled.

Despite the high precision production technique of the CellDrums, it is especially challenging to select a six-piece batch of membranes whose base compliance do not exceed 3% deviation. For this purpose, the individual CellDrums were numbered with a code that include mechanical parameters. The parameters (thickness, compliance) are used in the final individual CellDrum product sheet in order to calculate the mechanical stress of cell and tissue layers. This is the only way to ensure comparable and repeatable measurement of lateral cell forces of cultured cell monolayers. Future work will focus on this topic in detail.

## 4. Conclusions

In this study, we introduce a production and functionalization method for an ultra-thin, chemically functionalized PDMS membrane suitable for cell culture, along with standardized quality control procedures to determine important mechanical characteristics. We investigated the dependency of membrane thickness to four parameters; PDMS volume, room temperature (°C), membrane deflection and compliance. No correlation was found between membrane thickness and stiffness in the thickness range of 3.28–3.98 µm. Once the thickness range was widened to 3.24–10.06 µm, significant changes on membrane mechanical properties were revealed. Further clear differences were seen in different ambient temperature and PDMS volume settings in function of membrane thickness. With our results, the hypotheses of other studies mentioned above, were supported. Mechanical properties of PDMS membranes highly depend on various environmental parameters such as curing temperature, room temperature, mixing ratio and membrane thickness. All in all, this study proves how even the smallest fluctuations can create significant differences in CellDrum membrane mechanical properties, showing the dire need of standardized quality management procedures. Without precise quality management of the membranes, no reproducible measurements of the change in compliance due to cell force change are possible. Therefore, it is absolutely necessary to test every single membrane with regard to its sensitivity and to enter this into the evaluation algorithm. Only with detailed quality control procedures can such thin membranes be used with real reliability for cellular force measurements and their changes with sensors. The quality management procedure must include these steps starting from production: (1) semi-automated membrane fabrication with constant PDMS volume at constant room temperature in a dust free environment, (2) heat curing in an enclosed space with stabilized temperature regulation, (3) transfer of the membrane onto the applicator, (4) surface functionalization, (5) measurements of mechanical properties in controlled environments (thickness, deflection, compliance), (6) sterilization and finally, (7) labeling of each single membrane with all the measured values as input information. Reproducible membrane thickness is achieved by automating the transfer of PDMS to the water and by stabilization and precision of the membrane production environment. This applies to both the membrane thickness and the highly reproducible mechanical preload of the membrane that is created when it is clamped onto the CellDrum applicator (Figure 1A). Automation of PDMS transfer provides more accurate pipette volume compared to manual pipetting. A closed environment further improves the precision of membrane fabrication by ensuring freedom from dust and a constant room temperature, as well as uniform heat distribution during membrane curing.

The CellDrum technology along with its functionalized membrane offers a softer and more flexible surface for cellular experiments, which can be used in combination with the corresponding membrane compliance measurement device CFA_refer_. The device allows the measurement of membrane compliance as a function of deflection and pressure. It can be further applied for a variety of studies such as wound healing, drug testing, gene expression as well as mechanically testing tissue engineering approaches. This membrane fabrication method could be used as an alternative to the commercially available soft cultivation grounds since they not only allow measurement of cellular forces but also the application of external mechanical forces onto the cellular layer thanks to its flexible ground.

The use of membranes defined with high mechanical precision, as demonstrated in this work, is not limited to the measurement of cell cultures only. In the future, we can imagine performing general material tests based on the membrane fabrication shown here. This could include industrial application in test procedures for example to quantify the curing characteristics of adhesives and dyes, to test the mechanical behavior of tough polymers, or to study even phase transitions in polymers. However, in order to obtain comparable and repeatable mechanical data of thin films with such thin membranes, multi-level quality management systems have to be applied, as presented in this paper.

## Figures and Tables

**Figure 1 polymers-14-02213-f001:**
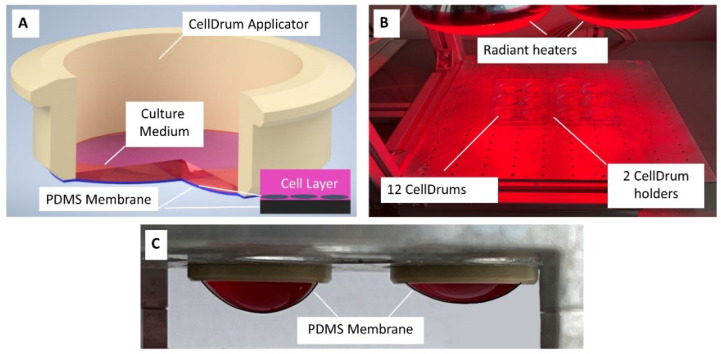
Scheme: PDMS membrane fabrication and testing setup. (**A**) Single CellDrum with PDMS membrane (black, enlarged inset), cells (gray, enlarged inset) with culture medium (magenta). (**B**) CellDrum membrane curing with heat lamps (**C**) Deflection of two membranes under different liquid loads with 2 mL (**left**) and 1 mL (**right**) cell culture medium.

**Figure 2 polymers-14-02213-f002:**
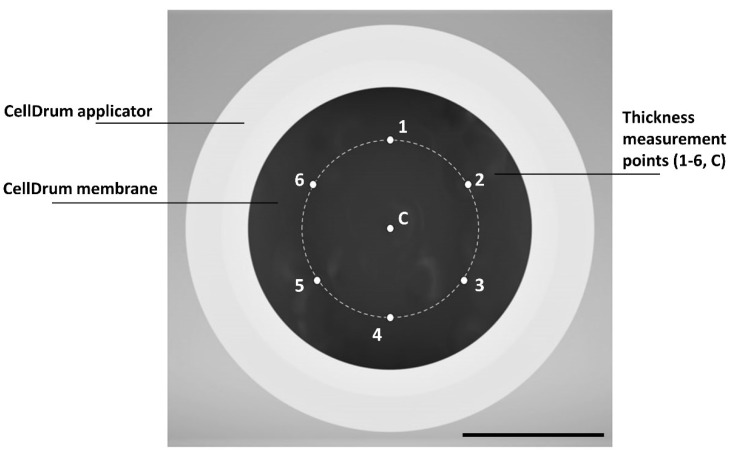
Positioning of thickness measuring points for CellDrum membranes (**top** view). Inner membrane diameter is 1.6 cm, and angular spacing of the data points on the measuring circle is 60°. Measurement points are numbered from 1–6 for eccentric measurements and C for central, respectively. The scale bar represents 8 mm in length.

**Figure 3 polymers-14-02213-f003:**
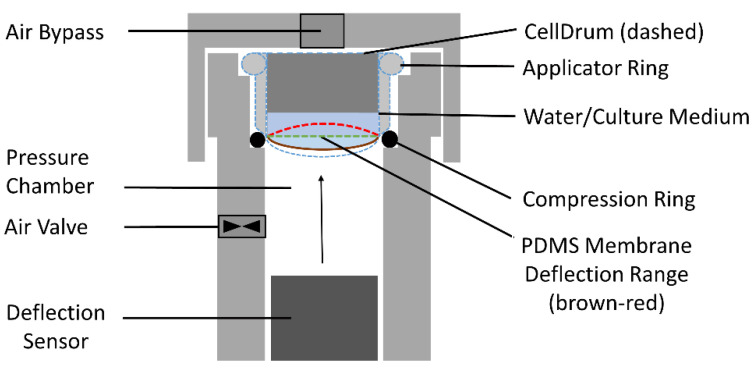
Schematic cross-section of the deflection measurement system. The contour of the CellDrum is shown dashed in blue. The PDMS membrane forms the lower part of the CellDrum. The figure indicates three membrane positions: Brown and red mark the lower and upper limits, respectively, of the membrane bending range used for the measurement. Green marks the flat membrane condition, i.e., the strain minimum. The membrane curvature is determined by the pressure in the pressure chamber, the membrane properties (PDMS membrane with attached cell culture layer) and the weight of the water/medium above the membrane in the CellDrum.

**Figure 4 polymers-14-02213-f004:**
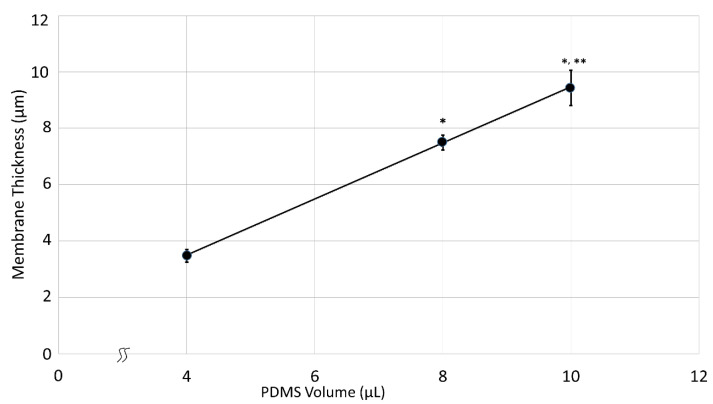
PDMS membrane thickness as a function of PDMS volume. Averages and standard deviations of all groups (*n* = 11 per group) are shown including the regression line. All groups showed a significant difference between each other within a 99% confidence interval (* *p* < 0.01 vs. 4 µL group, ** *p* < 0.01 vs. 8 µL group). r^2^ = 0.97.

**Figure 5 polymers-14-02213-f005:**
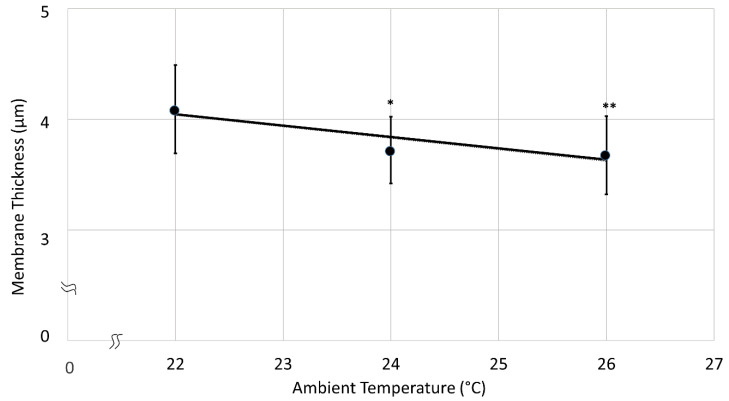
Membrane thickness as function of the environmental temperature. A one-way ANOVA test showed significant differences between the groups 1–2, 1–3; however, none between 2–3. (* *p* < 0.05 vs. Group 1, ** *p* < 0.01 vs. Group 1, r^2^ = 0.83).

**Figure 6 polymers-14-02213-f006:**
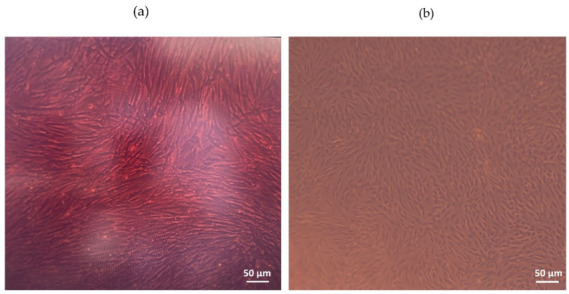
Images of cell monolayers cultured on the CellDrum PDMS membrane under phase-contrast microscope (**a**) Human arterial smooth muscle cells (haSMC), and (**b**) Human Dermal Fibroblasts-Adult (nHDFa).

**Figure 7 polymers-14-02213-f007:**
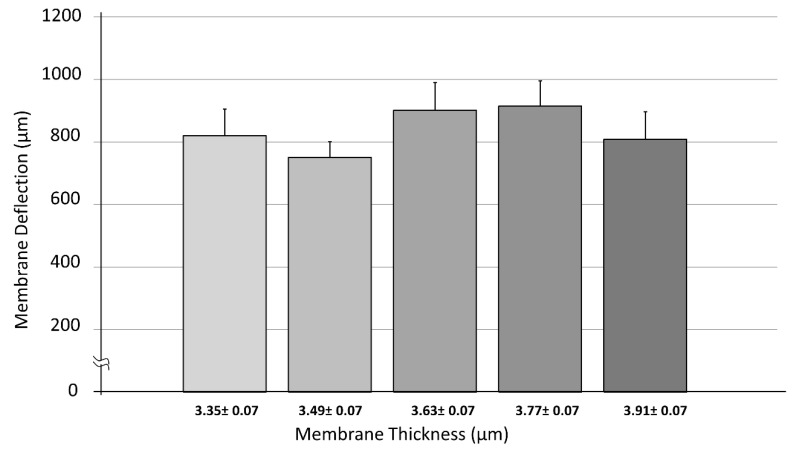
Dependency of membrane deflection vs. PDMS membrane thickness within a range of 3.28–3.98 µm. Data were tested for normality (*n* = 35) using Shapiro–Wilk test. Five thickness groups were evaluated, and their average deflection values were calculated. The “±” symbol defines the thickness range and is not standard deviation, error bars represent standard error of mean (SEM). One-way ANOVA test did not show significant differences between groups (α = 0.05).

**Figure 8 polymers-14-02213-f008:**
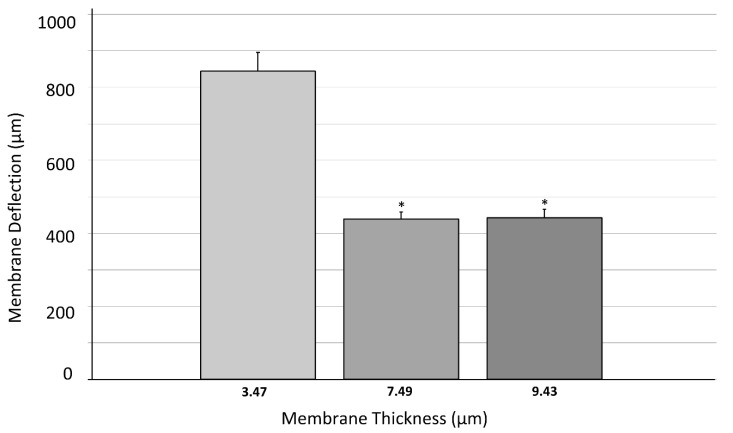
Dependency of membrane deflection on PDMS membrane thickness (thickness range = 3.24–10.06 µm). Data were checked for normality using Shapiro–Wilk test (*n* = 11 per group). Three different PDMS volumes were used to produce membranes with different thicknesses and their average deflection values were calculated. Error bars represent standard error of mean (SEM) for each group. One-way ANOVA test showed significant difference between groups 1–2 and 1–3 (4 and 8 µL and 4 and 10 µL PDMS volume group) with α = 0.01. Groups 2–3 (8–10 µL PDMS volume) showed no difference through ANOVA. (* *p* < 0.01 vs. Group 1).

**Figure 9 polymers-14-02213-f009:**
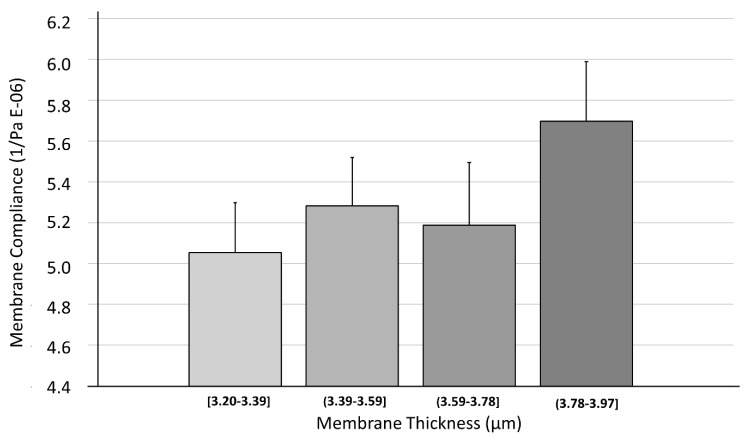
Dependency of membrane compliance on PDMS membrane thickness. Data were checked for normality (*n* = 30) using Shapiro–Wilk test. With the help of a histogram, four thickness groups were separated and their average deflection values were calculated for the graph. Error bars represent standard error of mean (SEM) for each group. One-way ANOVA test did not show any difference between the different thickness groups at α = 0.05.

**Table 1 polymers-14-02213-t001:** Experimental conditions for membrane thickness testing. Two parameters were tested.

Experimental Conditions	PDMS Volume (µL)	Room Temperature (°C)	Tested Parameter
1	4.00	22.00	PDMS volume
2	8.00	22.00	
3	10.00	22.00	
4	4.15	22.00	Ambient Temperature
5	4.15	24.00	
6	4.15	26.00	

**Table 2 polymers-14-02213-t002:** Measured membrane thickness values at various PDMS volumes. Numbers in parentheses represent the “calculated” values for the chosen volumes.

PDMS Volume (µL)	Average Thickness (µm)	Standard Deviation (µm)
4.00	3.47	0.23
8.00	7.49 (6.94) ^1^	0.25
10.00	9.43 (8.68) ^2^	0.63

^1,2^ The average thickness value for 4 µL group was used as a standard to calculate the theoretical/expected thickness values for membranes produced with 8 and 10 µL PDMS volume.

**Table 3 polymers-14-02213-t003:** Measured thickness values of membranes produced in different room temperatures.

Temperature (°C)	Average Membrane Thickness (µm)	Standard Deviation (µm)
22.00	4.09	0.40
24.00	3.72	0.30
26.00	3.68	0.35

**Table 4 polymers-14-02213-t004:** Measured average deflection values of membranes with different thickness ranges.

Membrane Thickness (µm)	Average Membrane Deflection (µm)	Standard Error of Mean (µm)
(3.28–3.42)	820.33	84.54
(3.42–3.56)	751.22	48.76
(3.56–3.70)	901.67	88.07
(3.70–3.84)	914.92	80.36
(3.84–3.98)	808.73	87.56

**Table 5 polymers-14-02213-t005:** Measured average deflection values of membranes with average membrane thicknesses ± standard deviation in µm.

Avg. Membrane Thickness (µm)	Avg. Membrane Deflection (µm)	Standard Error of Mean (µm)
3.47 ± 0.23	844.10	51.53
7.49 ± 0.25	439.27	19.20
9.43 ± 0.63	442.64	23.74

**Table 6 polymers-14-02213-t006:** Measured compliance values of membranes with different thickness ranges.

Membrane Thickness (µm)	Average Membrane Compliance (1/Pa)	Standard Error of Mean (1/Pa)
[3.20–3.39]	5.05 × 10^−6^	2.45 × 10^−7^
(3.39–3.59]	5.28 × 10^−6^	2.37 × 10^−7^
(3.59–3.78]	5.19 × 10^−6^	3.08 × 10^−7^
(3.78–3.97]	5.70 × 10^−6^	2.92 × 10^−7^

## Data Availability

The data presented in this study are available on request from the first author.

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
