# Peer review of "Bio-Functionalized Ultra-Thin, Large-Area and Waterproof Silicone Membranes for Biomechanical Cellular Loading and Compliance Experiments"

_polymers, 2022, doi:10.3390/polym14112213_

Round 1

Reviewer 1 Report

Caption of Fig 1. is confusing, where are cells (enlarged inset) in Fig 1b, more elaboration of Fig 1c is needed. What is the info revealed from this Fig 1c?

More discussion of the importance to fabricate the ultra-thin membranes is needed in the introduction.

Scale bar for Fig 2. and Fig 6. is needed.

Cross-section SEM images of  PDMS membrane is highly suggested for the verification of the thickness measurement.

Photos of the intact membranes are needed.

More discussion and supportive data are needed to highlight how to fabricate the membrane with reproducible thickness.

Reviewer 2 Report

The manuscript entitled "Bio-functionalized ultra-thin, large-area and waterproof Silicone membranes for biomechanical cellular loading and compliance experiments" is a nice contribution in the field of silicones for biomedical applications.

Overall, the paper is well written and the novelty and originality of methods are also confirmed. Sufficient data are presented in the Introduction section referring to the silicone as membranes highlighting the main characteristics for such applications: mechanical properties, surface biocompatibilization, thickness, flexibility, etc. The originality of the work consisted in the development of a membrane production procedure coupled with standard quality protocols to prepare well defined membranes.

In the Materials and Methods section, the Materials part is missing and this must be added. In Membrane fabrication section the authors mentioned a PDMS elastomer kit. They should add some details regarding the elastomer characteristics (molar mass, viscosity, functionality, etc ). Did authors try to produce similar elastomers in the lab? Did these ones be applied in this membrane protocols? The membrane functionalization procedure was realized by authors? or is a general procedure to optimize PDMS. I consider that some details about PDMS functionality should be added in this section. The same observation is for biological functionalization. If these procedures were made by authors the confirmation analysis supporting these structural changes on surface must be added.

Overall, the paper is of interest and suitable for publication after provided the above mentioned information. I recommend the acceptance after minor revision.

Round 2

Reviewer 1 Report

The authors addressed the comments properly, it can be accepted in present form.